# Stable Carbon and Sulfur Isotope Characteristics of Stream Water in a Typical Karst Small Catchment, Southwest China

**Yang Tang** [1],* and **Ruiyin Han** [2]

[1] Institute of Geochemistry, Chinese Academy of Sciences, Guiyang 550002, China
[2] The College of Resources and Environmental Engineering, Guizhou University, Guiyang 550025, China; ruiyinhan@163.com
* Correspondence: tangyang@mail.gyig.ac.cn; Tel.: +86-0851-85891334

**Abstract:** Surface water samples from the Maolan National Natural Reserved Park (MNNRP) were analyzed from Sept. 2013 to June 2014, for major ion concentrations ($K^+$, $Na^+$, $Ca^{2+}$, $Mg^{2+}$, $Cl^-$, $SO_4^{2-}$, $HCO_3^-$), $\delta^{13}C$-DIC and $\delta^{34}S$-$SO4^{2-}$ to quantify the sources of solutes and chemical weathering. The results show that $HCO_3^-$ and $SO_4^{2-}$ are the main anions in Banzhai watershed, which account for 86.2 and 10.4% of the total anion equivalent, respectively. While $Ca^{2+}$ and $Mg^{2+}$ account for 76.9 and 20.5%, respectively. Considerable $Mg^{2+}$ in stream water indicates that it may be affected by dolomite weathering. stream water samples present the $\delta^{13}C$-DIC values in the range of $-16.9‰ \sim -10.8‰$ (mean value was $-13.9‰$), which were lower than that of the groundwater. The $\delta^{34}S$-$SO_4^{2-}$ values ranged from $-15.2‰$ to $1.7‰$ (mean value was $-4.4‰$). There was a negative correlation between $HCO_3^-$ content and $\delta^{13}C$ value, implying the result of the interaction of temperature and precipitation intensity in different seasons. The significant positive correlation between $SO_4^{2-}$ content and $\delta^{13}C$-DIC indicates that $H_2SO_4$ may be involved in the weathering process of carbonate rocks in small watershed scale. The content of $SO_4^{2-}$ in a school sample site was much higher than that of other sample sites for the interference from human sources. The $\delta^{34}S$ values show that the average $\delta^{34}S$-$SO_4^{2-}$ in most sites is close to the $\delta^{34}S$ isotopic values of Guizhou coal and rain, indicating that they may be affected by local coal.

**Keywords:** C isotope; S isotope; water chemistry; karst terrain; Southwest China

## 1. Introduction

Previous studies showed that the sources of stream solutes come from natural processes and anthropogenic inputs. In these processes, the sources of various solutes need to be further improved [1,2]. Chemical weathering is the main sources of the stream solutes and is also the most important sinks of atmospheric $CO_2$ in watersheds [1,3,4]. In general, the dissolved inorganic carbon (DIC) in stream water come from soil $CO_2$, dissolution of carbonate minerals and atmospheric $CO_2$ [5]. C isotopes of the DIC are widely used as a tracer to discriminate and quantify carbon sources and processes involved in the stream carbon cycle [6–9]. However, the variations of $\delta^{13}C$ in DIC remain quite difficult to interpret despite carbon isotopes of carbonate minerals and of soil $CO_2$ are distinctive enough, because streaming respiration and isotopic equilibration with atmospheric $CO_2$ [10,11]. Although $CO_2$ is the main agent of chemical weathering of carbonate rocks, other acidic substances, including sulfuric acid, nitric acid and so on, are involved in the process of chemical weathering in various ways. These acidic substances are not only produced in natural processes such as volcanoes, lightning, atmospheric photochemical reactions, natural oxidation of minerals, but also from human activities dominated by fossil fuel combustion. With the process of industrialization and urbanization, the more acidic substances released in the process of human activities cannot be ignored [12]. Chemical weathering can be influenced by human perturbations such as sulfuric acid ($H_2SO_4$) and nitric acid ($HNO_3$), and these acids can also accelerate chemical weathering rate [13–16]. Moreover, it is feasible

to distinguish the sources of stream $SO_4^2$ by using the S isotope [13,17,18]. Therefore, in combine with the C and S isotopes, the geochemical processes of the chemical weathering and its environmental effect can be examined.

The carbonate rocks are widely distributed in southwest China, where the total area of carbonate outcrops is about 540,000 $km^2$, which is one of the largest karst areas in the world [19]. However, there are few studies about the temporal variability of carbon biogeochemical cycles and effects of carbonate weathering in the small karst catchment scale. As a typical karst small catchment area, the Maolan National Natural Reserved Park (MNNRP) is essentially mono-lithology, and there is little isotope variation from the weathering of different rock types [20]. Therefore, MNNNRP is an ideal site to study the effects of carbonate weathering processes by carbonic acid and potential sulfur acid.

As the typical karst catchment in southwest China, much research has been conducted to examine the compositions of rainwater [20,21], groundwater [20] and soil [22,23] in MNNRP area. However, little research of the multi-isotopic composition of stream water has been investigated systematically up to now. This study conducts stream water analyses including major ions, $\delta^{13}$C-DIC and $\delta^{34}$S-SO$_4$ and aims to investigate (1) analysis of the spatial-temporal distribution of the major ions, $\delta^{13}$C-DIC and $\delta^{34}$S-SO$_4$ in small watershed scale and its influencing factors; (2) the C and S isotope coupling mechanism in the weathering process of carbonate rocks, exploring the significance of the $\delta^{13}$C-DIC and $\delta^{34}$S-SO$_4$ signals during the identification of sources of solutes. The results would greatly help manage and protect the water resources and provide water resources a guarantee from the small karst watershed scale.

## 2. Materials and Methods

### 2.1. Study Site

The Maolan National Natural Reserved Park (MNNRP) is famous for its dense virgin evergreen forests growing on peak cluster karst. This park located in the subtropical monsoon region, and the precipitation is concentrated into the high-sun season. The sampling site is situated in the MNNRP, which is in the southeast of Guizhou province, southwest China. The carbonate of middle- and lower-Carboniferous ages are widely distributed in this study area. The karst geomorphology, together with the forest growing on the base rock in the habitat, forms a unique natural complexity of karst forest in the subtropical region. The lithology of Banzhai watershed system is middle Carboniferous limestone and dolomite. The occurrence of the strata is gentle. There are lower Carboniferous marl, shale and siliceous rock strata only near the downstream of the watershed.

### 2.2. Sampling

The field surveys were conducted from September 2013 to June 2014. Four samples were collected from each sampling site according to spring (March), summer (June), autumn (September) and winter (December). A total of 24 steam water samples were collected in a small karst catchment called Banzhai in the Maolan National Natural Reserved Park (Figure 1). Six sampling points have been set up in this small watershed:

- School (25°14′05″ N, 108°01′51″ E, H 542 m.a.s.l), well.
- Liming (25°13′28″ N, 108°01′25″ E, H 554 m.a.s.l), spring.
- Village (25°13′53″ N, 108°01′02″ E, H 602 m.a.s.l), spring.
- Upstream (25°13′07″ N, 108°00′24″ E, H 560 m.a.s.l), river water.
- Midstream (25°13′21″ N, 108°00′52″ E, H 539 m.a.s.l), river water.
- Downstream (25°13′48″ N, 108°01′34″ E, H 529 m.a.s.l), river water.

The water samples were collected at a depth about 20 cm from the stream on the bank with pre-cleaned 1.5-L high-density polyethylene (HDPE) bottles. The samples for $\delta^{13}$C-DIC analysis were filtered through a pre-cleaned 0.45 µm cellulose acetate filter membrane, stored in the polyethylene bottles with air-tight caps, and $HgCl_2$ was added to prevent biological activity. The stream samples for $\delta^{34}$S-SO$_4$ and ions concentration analysis were filtered through a pre-cleaned 0.22 µm membrane (Millipore), the samples

for anions measurement were directly kept in HDPE bottles, while the samples for cations and S isotopic detection were kept in the HDPE bottles after acidification with the high-purity HCl to pH < 2. All the containers were sealed and kept refrigerated at about 4 °C until analyses.

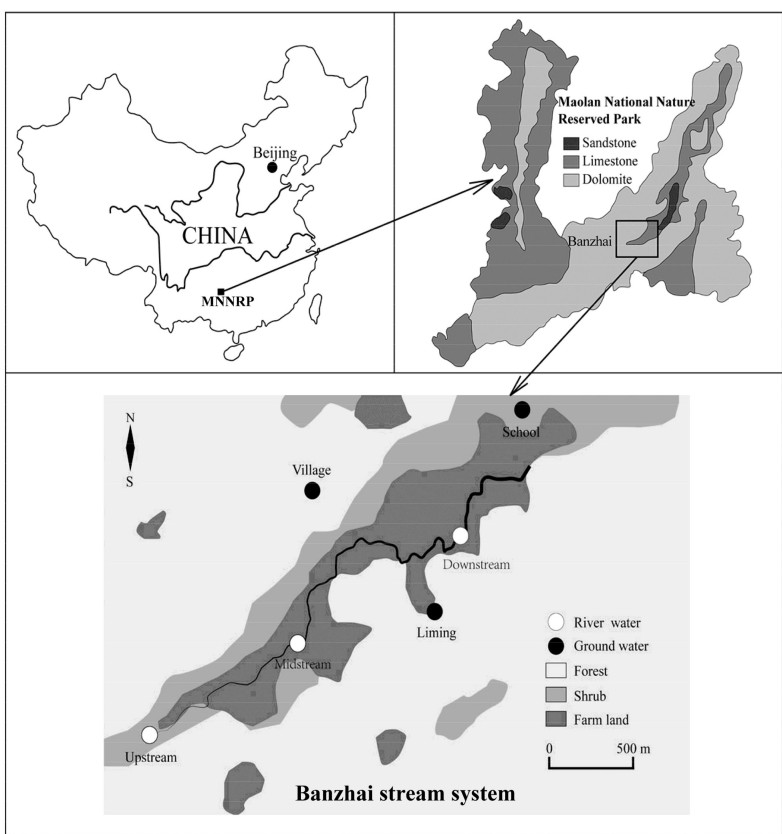

**Figure 1.** Sketch map showing the geology and land use of study area.

## 2.3. Analytical Procedure

EC, pH and other physical–chemical parameters of stream water samples were measured in the field by 6920-YSI water quality analyzer. The $HCO_3^-$ concentrations were titrated by HCl in situ. The cations ($Na^+$, $K^+$, $Ca^{2+}$, $Mg^{2+}$) and anions ($Cl^-$, $NO_3^-$, $SO_4^{2-}$) were determined by DIONEX ICS-900 ion chromatograph in Guizou University. The normalized ion charge ($[TZ^+-TZ^-]/TZ^-$) of most samples is less than ±5%, replicate samples and laboratory mixing standard solutions were employed to assess the accuracy of all the analysis, the relative standard deviations of all the analysis were within ±5%.

The pretreatment of the carbon stable isotope measurement followed the modified method [24] and was employed many times in our previous studies [20]. In short, each ~10 mL sample was injected into glass bottles with by the syringes, 1 mL 85% phosphoric acid and magnetic stir bars were set in the bottles in advance. The dissolved carbon species were converted as $CO_2$ by the acid, then the $CO_2$ was extracted into a vacuum line and pass through the $N_2$ cooled ethanol trap to separated $H_2O$; finally, the $CO_2$ was transferred into a tube under cryogenical condition for the next isotope measurement. The C isotope ratios of DIC were measured by the Finigan MAT 252 mass spectrometer and reported by the δ notation relative to PDB in per mil, as follows:

$$\delta^{13}C(‰) = (\frac{R_{sample}}{R_{PDB}} - 1) \times 1000 \qquad (1)$$

The pretreatment method of S stable isotope followed previous studies [16,18]. In brief, the dissolved $SO_4^{2-}$ in samples was converted as barium sulfate precipitation by

using the 10% barium chloride solutions ($BaCl_2$); after resting 48 h, the mixture was filtered through 0.22 μm membrane filters. The precipitation ($BaSO_4$) was calcined under 800 °C for about 40 min. The S isotopic compositions of $BaSO_4$ were measured by Finnigan Delta-C isotope IRMS and were reported by the δ notation relative to Vienna-Canyon Diablo Troilite (VCDT) in per mil as follow:

$$\delta^{34}S(‰) = \left(\frac{R_{sample}}{R_{PDB}} - 1\right) \times 1000 \tag{2}$$

The standards NBS19, CO9, GBW04416, GBW04417 for C isotope and NBS 127, IAEA-SO-5($BaSO_4$) for S isotope were measured to check the accuracy of all the analysis; the results were within the recommended values, the 2SD values were within ±0.02‰.

## 3. Results

### 3.1. Variations in Physico-Chemical Data of Water Samples

Major ion concentrations, $\delta^{13}C$-DIC and $\delta^{34}S$-$SO_4^{2-}$ ratios of the water samples are given in Table 1. The water temperature ranged from 12.6 °C in winter to 22.8 °C in summer. The mean pH value was 7.4 (ranging from 6.7 to 8.1) indicating the water samples were slightly alkaline. The normalized ion charge ($[TZ^+ - TZ^-]/TZ^-$) was less than ±10%; the slight imbalance may result from the organic complex matters [25].

**Table 1.** Hydro-chemical composition and $\delta^{13}C$-DIC and $\delta^{34}S$-$SO_4^{2-}$ ratios of water in Banzhai watershed.

| Site | Season | Temp | pH | EC | TDS | $HCO_3^-$ | $Cl^-$ | $NO_3^-$ | $SO_4^{2-}$ | $Na^+$ | $k^+$ | $Mg^{2+}$ | $Ca^{2+}$ | $\delta^{13}C$ | $\delta^{34}S$ |
|------|--------|------|----|----|----|----|----|----|----|----|----|----|----|----|----|
|  |  | °C |  | μS.cm$^{-2}$ | mg.L$^{-1}$ | meq.L$^{-1}$ | meq.L$^{-1}$ | meq.L$^{-1}$ | meq.L$^{-1}$ | meq.L$^{-1}$ | meq.L$^{-1}$ | meq.L$^{-1}$ | meq.L$^{-1}$ | ‰ | ‰ |
| Up | Spr. | 15.8 | 8.13 | 501 | 235 | 3.28 | 0.04 | 0.10 | 0.24 | 0.02 | 0.01 | 1.17 | 2.66 | −14.83 | −1.50 |
| Up | Sum. | 21.9 | 7.52 | 496 | 233 | 3.82 | 0.02 | 0.07 | 0.18 | 0.02 | 0.01 | 1.19 | 2.77 | −14.99 | 1.20 |
| Up | Fal. | 20.9 | 7.78 | 456 | 214 | 3.40 | 0.04 | 0.09 | 0.20 | 0.02 | 0.02 | 1.35 | 2.15 | −16.87 | −0.78 |
| Up | Win. | 17.3 | 7.69 | 487 | 229 | 3.79 | 0.04 | 0.10 | 0.21 | 0.03 | 0.02 | 1.38 | 2.89 | −15.42 | −1.01 |
| Mid | Spr. | 13.7 | 7.71 | 466 | 219 | 3.39 | 0.04 | 0.10 | 0.26 | 0.02 | 0.01 | 1.13 | 2.43 | −13.13 | −2.17 |
| Mid | Sum. | 22.1 | 7.46 | 451 | 212 | 3.58 | 0.02 | 0.07 | 0.17 | 0.02 | 0.01 | 1.14 | 2.78 | −15.00 | −0.56 |
| Mid | Fal. | 22.6 | 8.02 | 409 | 192 | 3.36 | 0.04 | 0.06 | 0.21 | 0.02 | 0.02 | 1.33 | 2.45 | −13.68 | −1.01 |
| Mid | Win. | 13.2 | 7.95 | 423 | 199 | 3.97 | 0.04 | 0.07 | 0.23 | 0.03 | 0.02 | 1.33 | 2.72 | −13.13 | −2.13 |
| Down | Spr. | 15.3 | 7.36 | 486 | 228 | 3.44 | 0.06 | 0.06 | 0.36 | 0.05 | 0.02 | 0.98 | 2.78 | −12.30 | −3.63 |
| Down | Sum. | 22.8 | 6.98 | 479 | 225 | 3.77 | 0.02 | 0.06 | 0.20 | 0.02 | 0.01 | 0.99 | 2.94 | −14.91 | −1.72 |
| Down | Fal. | 22.6 | 7.74 | 417 | 196 | 3.58 | 0.05 | 0.03 | 0.23 | 0.03 | 0.02 | 1.28 | 2.68 | −14.01 | −1.78 |
| Down | Win. | 12.8 | 7.25 | 438 | 206 | 3.51 | 0.05 | 0.02 | 0.30 | 0.05 | 0.02 | 1.21 | 2.70 | −12.04 | −2.29 |
| school | Spr. | 12.7 | 6.65 | 740 | 348 | 3.88 | 0.40 | 0.33 | 2.04 | 0.46 | 0.24 | 0.59 | 5.18 | −10.80 | −11.83 |
| school | Sum. | 22.3 | 7.11 | 695 | 327 | 3.70 | 0.13 | 0.13 | 1.65 | 0.16 | 0.10 | 0.40 | 5.16 | −14.51 | −15.21 |
| school | Fal. | 22.1 | 6.87 | 638 | 300 | 3.31 | 0.31 | 0.17 | 1.52 | 0.45 | 0.18 | 0.55 | 4.00 | −12.84 | −6.22 |
| school | Win. | 12.6 | 7.03 | 723 | 340 | 4.62 | 0.31 | 0.17 | 1.52 | 0.56 | 0.20 | 0.62 | 5.36 | −11.07 | −11.23 |
| liming | Spr. | 14.8 | 7.34 | 509 | 239 | 3.39 | 0.04 | 0.08 | 0.43 | 0.10 | 0.02 | 0.32 | 3.40 | −13.73 | −9.92 |
| liming | Sum. | 21.9 | 7.19 | 542 | 255 | 3.96 | 0.03 | 0.04 | 0.28 | 0.03 | 0.02 | 0.33 | 4.06 | −16.11 | −6.97 |
| liming | Fal. | 20.4 | 7.41 | 451 | 212 | 4.03 | 0.02 | 0.03 | 0.38 | 0.04 | 0.01 | 0.42 | 3.87 | −14.87 | −8.98 |
| liming | Win. | 16.9 | 7.25 | 521 | 245 | 3.76 | 0.03 | 0.05 | 0.46 | 0.06 | 0.02 | 0.42 | 3.65 | −13.42 | −10.61 |
| village | Spr. | 16.1 | 7.35 | 563 | 265 | 2.85 | 0.05 | 0.05 | 0.37 | 0.02 | 0.01 | 0.37 | 2.83 | −12.32 | −3.88 |
| village | Sum. | 22.4 | 7.17 | 525 | 247 | 4.40 | 0.03 | 0.03 | 0.23 | 0.03 | 0.00 | 0.51 | 4.03 | −16.16 | 1.70 |
| village | Fal. | 21.5 | 7.12 | 511 | 240 | 3.78 | 0.03 | 0.12 | 0.24 | 0.04 | 0.00 | 0.59 | 3.68 | −13.96 | −1.35 |
| village | Win. | 15.9 | 7.08 | 485 | 228 | 3.99 | 0.03 | 0.12 | 0.32 | 0.02 | 0.00 | 0.60 | 3.74 | −14.05 | −2.62 |

EC: Electrical conductivity; TDS: Total dissolved solids.

The relative proportions of major ions are showed in the piper diagrams (Figure 2). The order of the cations is $Ca^{2+} > Mg^{2+} > Na^+ + K^+$. The $Ca^{2+}$ is the dominant cations in water samples and constitutes more than 60% of the cations. The order of the anions is $HCO_3^- > SO_4^2 > Cl^-$; bicarbonate ($HCO_3^-$) was the dominant anion and contributes about more than 50% of the total anions in this watershed.

### 3.2. Temporal and Spatial Variation of C and S Isotope Compositions

The concentration of dissolved inorganic carbon is the sum of the inorganic carbon species in a given solution.

$$DIC = [HCO_3^*] + [HCO_3^-] + [CO_3^{2-}]$$

$$[HCO_3^*] = [CO_2(aq)] + [H_2CO_3(aq)]$$

The speciation of DIC is pH controlled, with increasing proportions of $CO_2$(aq) as pH decreases [26]. In most of the water samples, bicarbonate was the main component of DIC, accounting for about more than 90% of DIC.

From Table 1, it is can be seen that the $\delta^{13}$C-DIC of most samples present the similar values with a narrow range of −10.8 to −16.9‰; the mean value was −13.9‰. In contrast, the $\delta^{34}$S−$SO_4{}^{2-}$ exhibited large variation, ranging from −15.2‰ to 1.7‰ with the mean value of −4.4‰.

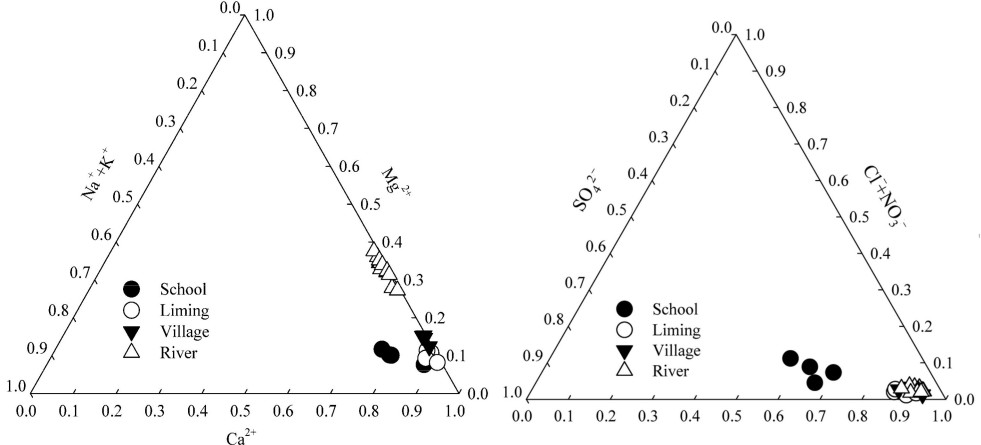

**Figure 2.** Piper diagrams showing chemical composition of major cations and anions in water of Banzhai.

## 4. Discussion

Carbonate dissolution is important in controlling stream water chemistry, as illustrated by the Ca–HCO3 dominance of our results (Figure 2). High $HCO_3{}^-$ and pH are typical for the stream waters that drain carbonate terrain. This is consistent with the geology of the small catchment being dominated by carbonate rocks ($CaCO_3$ and $CaMg(CO_3)_2$) [27].

### 4.1. Seasonal and Spatial Variations of the Solutes

Mean pH values of Banzhai water samples was 7.4, which is consistent with similar to the average pH value (7.8) of the main rivers of Guangxi province in this study area and is consistent with the alkalescency of natural water in carbonate area [28]. It can be seen that the TDS value in wet seasons (summer and fall) is lower than those in dry season (winter and spring) (Table 1). The TDS of groundwater in Banzhai watershed is 25.3% higher than that of surface water. The contents of solutes in the stream has the lowest TDS (192 mg.L$^{-1}$) in fall, which suggests that local rain events may dilute the contents of solutes in the catchment. Additionally, the highest TDS (235 mg.L$^{-1}$) suggested that the water is more likely to experience a longer water rock interaction process. This is slightly different from Wujiang river basin in Guizhou karst area, and the main stream of Wujiang river is about 250 km to the north of Banzhai; the TDS of Wujiang river basin in rainy season is slightly higher than that in dry season, and the TDS (237–434 mg.L$^{-1}$) of the Wujiang is higher than that of Banzhai (192–348 mg.L$^{-1}$) [29]. This may be due to the relatively low vegetation coverage, more agricultural activities and stronger leaching of carbonate rocks in Wujiang river basin. The population density of Banzhai is about 35 p.km$^{-2}$, and that of Wujiang river basin is about 260 p.km$^{-2}$.

The triangles of anions and anions can be used to analyze the chemical composition sources of different end members, and to distinguish the influence of weathering types of rocks on the ion components of water (Figure 2). All the bedrock in the study area is carbonate rock, so the effect of silicate on water solute can be ignored [1]. If $CO_2$ is involved in the weathering of carbonate rocks in the anion triangles, the projection point should be close to $HCO_3{}^-$ side; if the carbonate rocks are completely weathered by $H_2SO_4$, the projection should be located in the middle of $HCO_3{}^-$ and $SO_4{}^{2-}$ equivalent lines on

the anion triangle [25,27]. In the anion triangle diagram of water in Banzhai watershed (Figure 2), except for the school sample site, the rest of the sites are close to the $HCO_3^-$ end, indicating that the water of the Banzhai watershed is mainly $CO_2$ involved in the weathering process of carbonate rocks [27,30]. However, the groundwater of the school sample site also deviates to the side of $SO_4^{2-}$ equivalent line, which indicates that there may be additional $SO_4^{2-}$ sources in the water flow area; the groundwater outlet of the school site is about 2 m deep from the surface, and the human activities around this site are frequent, so the surface water may polluted by human activities, but it may also be related to the underground lithology of the water flow. The water of the Banzhai watershed can be divided into three types according to the cation triangle diagram. The groundwater of liming and village are the same type, almost all of them are only $Ca^{2+}$ ions, indicating that limestone weathering is the main process in the stratum where the groundwater flows [31]. The cations in the stream water of Banzhai are $Ca^{2+}$ + $Mg^{2+}$ mixed, which indicates that the water may be affected by the weathering of both limestone and dolomite. While the water cation in the school sample site is mainly $Ca^{2+}$ type, containing part of $Na^+$ and $K^+$, and its $SO_4^{2-}$ content is much higher than that of other sample sites. Therefore, the water in this site may be mixed with anthropogenic sources (such as fertilizer, domestic wastewater, etc.) [20]. The ion composition characteristics of water in Banzhai watershed are similar to those in the main stream of Beipanjiang river [32], both Beipanjiang river and Banzhai river belong to the Pearl river system, and the main stream of Beipanjiang is 200 km west of Banzhai. The difference is that the content of $Na^+$ + $K^+$ and $Cl^-$ and $SO_4^{2-}$ in the water of the Beipanjiang is slightly higher. This may be due to the fact that almost all the solutes in Banzhai water come from the natural weathering of carbonate rocks, although the acid ions in local precipitation may be related to agricultural activities [21].

### 4.2. Carbon Evolution and Controlling Factor

Chemical weathering of carbonate and silicate is very important for the carbon cycling [16,33]. The main ion in the water of Banzhai watershed is $HCO_3^-$. In addition to special school site, $HCO_3^-$ can account for 85.9–93.8% of the total anion equivalent, which is also consistent with the anion characteristics of natural water in most karst area (Figure 2). As the main carrier of dissolved inorganic carbon (DIC), the main sources of $HCO_3^-$ in the basin may include $CO_2$ dissolved in soil water, $HCO_3^-$ released from the hydrolysis of carbonate minerals, dissolved $HCO_3^-$ in atmospheric precipitation, $CO_2$ exchanged into water-by-water air interface and $CO_2$ released by respiration of aquatic plants [7,8,34]. For small-scale basins, water air exchange and $CO_2$ from aquatic plants is relatively low, which can be negligible [7]. The $\delta^{13}C$ of $CO_2$ in the atmosphere is about $-8.0‰$, and the $\delta^{13}C$ value in soil is dependent on the vegetation types [22]. The average $\delta^{13}C$ value of soil organic matter dominated by C3 plants is $-26‰$, while that of soil organic matter dominated by C4 plants is $-12.0‰$ [5], coupled with microbial decomposition of organic matter and carbon isotope fractionation during plant respiration, the average $\delta^{13}C$ value of $CO_2$ released by soil respiration is about $-27.0‰$ [35,36]. The carbonate rocks in this study area belong to marine sediments, and the value of $\delta^{13}C$ is generally considered to be $0.0‰$ [16].

On the whole, the $\delta^{13}C$ values of $HCO_3^-$ in the Banzhai watershed varied from $-16.9$ to $-10.8‰$, with an average of $-13.9‰$ (Figure 3), which is very close to the $\delta^{13}C$ value ($-14.00‰$) of groundwater DIC formed by carbonate rock dissolved by soil $CO_2$ in the open karst system [30]. This indicates that $CO_2$ released by soil respiration may be an important source of DIC in this catchment [5,37]. However, considering the karst background of carbonate rocks in the study area, the $HCO_3^-$ in the water may be mixed from multiple sources [38]. The $\delta^{13}C$ values of $HCO_3^-$ in Banzhai watershed were different in different seasons, that is, the $\delta^{13}C$ value was lowest in summer (average $-15.3‰$) but highest in spring (average $-12.6‰$). This kind of C isotope variation may be related to the biological activities and rainfall in different seasons. In summer, the microbial activity was strengthened, the decomposition rate of soil organic matter was accelerated

and more rainwater was also beneficial to dissolution of soil $CO_2$ [24]. This is consistent with the research results in Xijiang stream and another small karst basin with frequent agricultural activities in the same region in Puding [38–40]. As shown in Figure 3, there is an obvious negative correlation between the content of $HCO_3^-$ and its $\delta^{13}C$ value in Banzhai watershed [6], that is, $\delta^{13}C$ values are decreased with the increase in $HCO_3^-$ content. This negative correlation may be related to the precipitation and temperature in different seasons. Summer is a rainy season; abundant precipitation makes it easier for $HCO_3^-$ in the soil to be washed into the stream, so the $\delta^{13}C$ value of stream water will be negative [7]; at the same time, the hot and humid environment in summer promotes the dissolution of soil water to carbonate rock and then produces a relatively higher DIC concentration under the unsaturated condition [33]. In winter and spring, the precipitation is less, the biological activity is weak; at this time, DIC in water is more likely to come from the hydrolysis of carbonate rocks, so the $\delta^{13}C$ value of DIC will be positive.

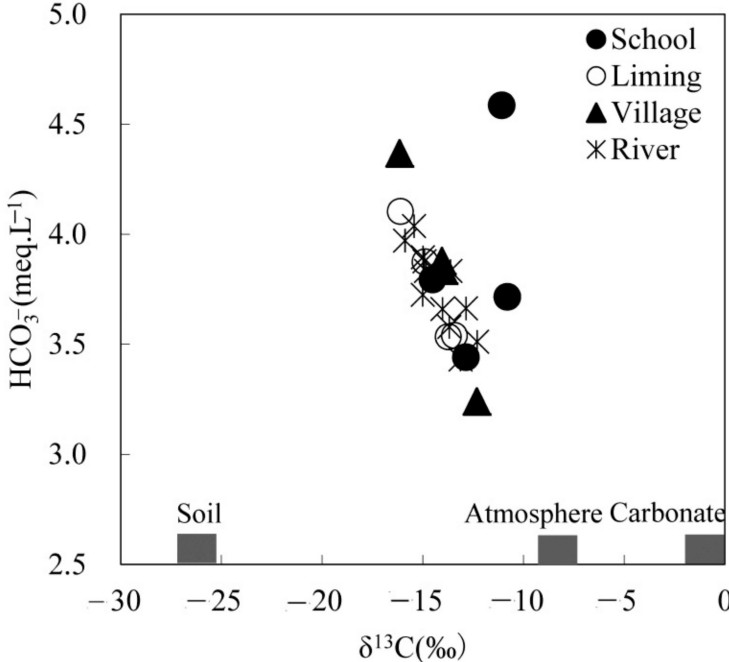

**Figure 3.** Effect of $HCO_3^-$ content on the $\delta^{13}C$ value in Banzhai water, the three main− potential sources of DIC are carbonate rock [16], atmosphere [22] and soil [35].

*4.3. Sulfur Isotope and Controlling Factor*

Previous studies suggested that $H_2SO_4$ accelerates the chemical weathering process [3,14,15]. Sulfur from different material sources has different sulfur isotopic composition. The natural sources of $SO_4^{2-}$ in surface water are mainly atmospheric deposition, groundwater, dissolution of sedimentary sulfate, oxidation of sulfide minerals and organic matter [13,17]. In Banzhai catchment, S stable isotope can be considered at the effective tool to distinguish the sources of sulfate ions. According to the conventional analysis and field investigation, the possible end members of sulfate sources in Banzhai basin include: 1. input from dry and wet atmospheric deposition; 2. dissolution of local sulfate minerals; 3. sulfide oxidation in underground coal seams; 4. released by local residents' activities [18,41,42].

The average $SO_4^{2-}$ equivalent content of each groundwater and stream water sample site of the Banzhai watershed is in the order of school > liming > village > stream (Table 1). The content of $SO_4^{2-}$ in the school site is much higher than that of other sampling sites, which indicates that the source of $SO_4^{2-}$ in school is quite different from that in other sampling sites. The results of $\delta^{34}S$ isotope test also showed that the average $\delta^{34}S$ isotopic values of the water samples of school and liming were −11.1 and −9.1‰ (Table 1), respectively, which was far lower than that of the village and stream water. It is suggested that the

sources of $SO_4^{2-}$ in water samples may be different. The similar seasonal variation in $\delta^{34}S$ isotopic values of the water samples from the other sampling sites were also observed, that is, the $\delta^{34}S$ isotopic values in spring were higher than those in other seasons.

The study area belongs to the typical marine monsoon climate, and its atmospheric precipitation is mainly from marine transport, and the $\delta^{34}S$ isotopic value in seawater is about +20.0‰ [43]. According to Xiao's research results in neighboring Guiyang, the averaged $\delta^{34}S$ isotopic value of rainwater ranged from −4.9 to +4.6‰ [44]; moreover, the sulfur in the heavy rain with positive $\delta^{34}S$ value is more likely to be marine origin. The $\delta^{34}S$ values of stream and village water are very close to the rainwater of Guiyang (Figure 4); therefore, $SO_4^{2-}$ in the water of these sites probably come from the infiltration of atmospheric precipitation and are less affected by other sulfur sources.

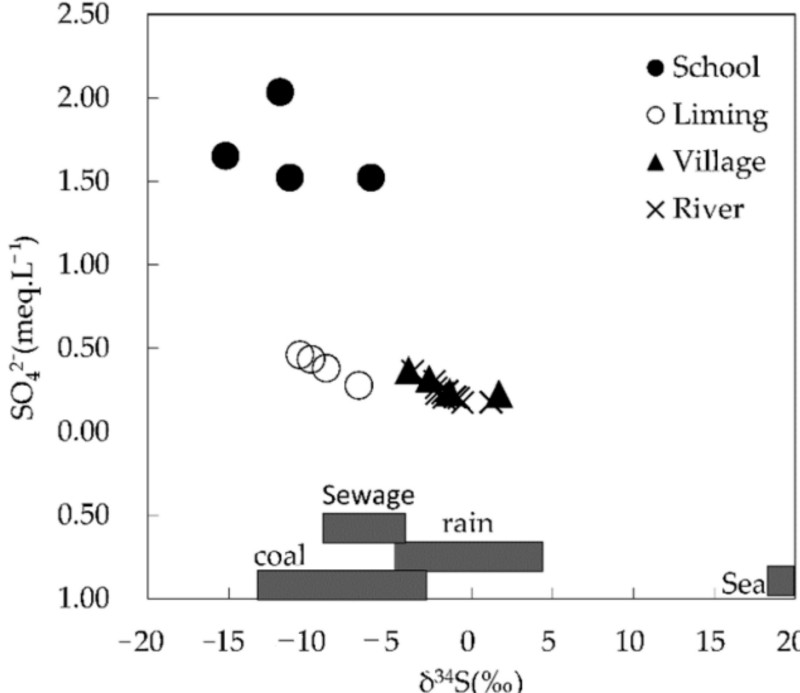

**Figure 4.** Relationship between $\delta^{34}S$ isotopic values and $H_2SO_4$ content in Banzhai water sample, the sulfur sources of the study area may be precipitation [44], coal [45], sewage [46] and seawater.

The $\delta^{34}S$ values in the water of the school and liming sites are relatively negative (−6.2–15.2‰), which are closer to the $\delta^{34}S$ of Guizhou coal (−12.5 to −2.5‰) [45]. The $\delta^{34}S$ values of school site are more negative, and the corresponding $SO_4^{2-}$ concentrations are very high, while the $SO_4^{2-}$ concentrations of liming site are closer to the relatively low concentration of stream water. Generally, the $SO_4^{2-}$ content in the runoff from rainwater is not too high, and the $\delta^{34}S$ value is positive, but if the water is mixed with additional sulfate (most of which are the products of combustion or oxidation of sulfur-containing minerals), the $SO_4^{2-}$ content will increase, while the $\delta^{34}S$ value will be negative. This negative correlation was obvious in all the samples except the school site (Figure 4). The $\delta^{34}S$ values of sewage and coal are partly coincident, although the $\delta^{34}S$ values of water in the school site are similar to those of coal; the high content of $SO_4^{2-}$ and the high content of $Na^+$ and $K^+$ suggest that the water is more likely to be mixed with sewage, while the $\delta^{34}S$ values of the liming site is also close to that of sewage and coal, but its relatively lower $Na^+$ and $K^+$ value indicates that it is less affected by sewage. The stream and village shows a range of $\delta^{34}S$ values similar to the local rainfall, indicating that the sulfur in these waters is more likely to come from precipitation.

### 4.4. Interaction between Sulfuric Acid and Carbon Isotope of DIC in Water

The effect of $NO_3^-$ on weathering is limited due to the low content of $NO_3^-$ (about 2% of the total anion equivalent) in the water of Banzhai watershed, while the observation results of local rainwater show that sulfate is the main cation [20,44]. Therefore, the influence of atmospheric input $SO_4^{2-}$ on weathering process is mainly considered in this discussion [16,18]. Although the $SO_4^{2-}$ in most water of Banzhai watershed is also at a low level (<0.5 meq $L^{-1}$), which indicates that exogenous acid may have little effect on the weathering of carbonate rocks in the Banzhai watershed [27]. However, there is an obvious positive correlation between $SO_4^{2-}$ and $\delta^{13}C$-DIC (Figure 5), that is, with the increase in $SO_4^{2-}$ concentrations, the $\delta^{13}C$-DIC values in the water is positive. The more negative the sulfur isotope of sulfate radical is, the more likely it comes from the oxidation of sulfur content in coal. At this time, the carbon isotope value of corresponding DIC is more positive, and these DIC are more likely to come from carbonate rock. Some studies have also confirmed that most of the s in the precipitation in this area comes from the combustion of local coal [45,47,48], this indicates that sulfur in coal may be an important driving force for carbonate rock weathering in the study area. This positive correlation may reveal that the weathering of carbonate rocks in Banzhai is affected by $H_2SO_4$. The process of $H_2SO_4$ participating in carbonate weathering is as follows:

$$2(Ca_{1-x}Mg_x)CO_3 + H_2SO_4 \rightarrow 2(1-x)Ca^{2+} + 2xMg^{2+} + 2HCO_3^- + SO_4^{2-}$$

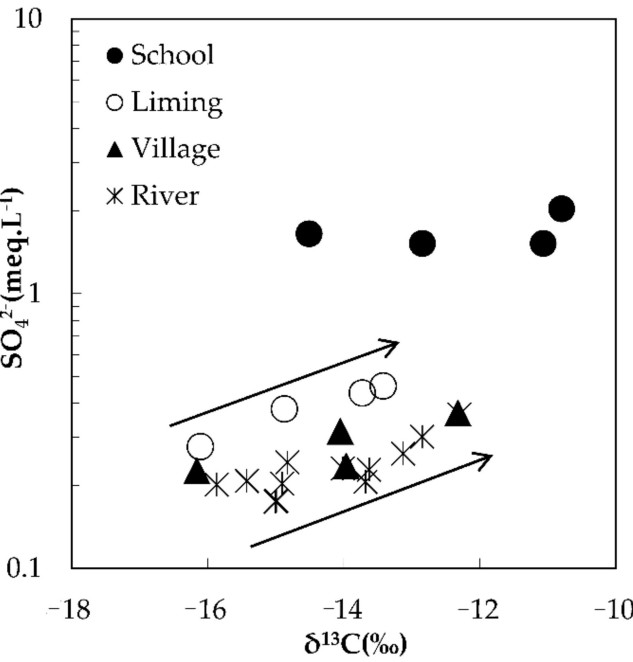

**Figure 5.** The variation trend of $SO_4^{2-}$ and $\delta^{13}C-DIC$ after $H_2SO_4$ participates in carbonate rock weathering process.

In this weathering process, 2 mol of $HCO_3^-$ will be generated by the complete reaction of 1 mol $H_2SO_4$, and all the 2 mol $HCO_3^-$ comes from the reacted carbonate rocks. Therefore, the $HCO_3^-$ generated during the weathering process of $H_2SO_4$ completely retains the carbon isotope characteristic values of carbonate rocks [14,49]. The carbonate rocks in Guizhou karst area belong to marine sediments, and their $\delta^{13}C$ values are generally considered to be 0.0‰, while those of biogenic origin are generally lower than −10.0‰ [38,40]. If $H_2SO_4$ is involved in the weathering and dissolution of carbonate rocks, the proportion of $HCO_3^-$ from carbonate rocks will increase, so the $\delta^{13}C$ value of DIC in water will gradually be positive to 0.0‰ [50]. In particular, the $SO_4^{2-}$ content of the school site was significantly higher than that of the other sampling sites, and the $\delta^{13}C$ value of DIC was not significantly different from that of other sampling sites, and the correlation between $SO_4^{2-}$ content

and $\delta^{13}$C value of DIC was not obvious (Figure 5). This indicates that the extent of $H_2SO_4$ participating in carbonate weathering of this site is consistent with that of other sampling sites. Most of the $SO_4^{2-}$ in domestic sewage has been neutralized, not directly related to the weathering of carbonate rocks, so there is no correlation between its concentration and C isotope value, the abnormally high content of $SO_4^{2-}$ in the water of the school site may be caused by the mixing of sewage discharged [51].

## 5. Conclusions

We have investigated the major concentrations, C isotope compositions and S isotope compositions of stream water in MNNRP, where carbonate rocks occupy the watershed. The water can be mainly characterized as $HCO_3$-Ca type according to the piper diagram. $HCO_3^-$ is the main species of DIC in the stream. The values of $\delta^{13}$C and $\delta^{34}$S in stream imply that human activities have little direct effect on the water evolution except the school site. Hence, the compositions of $\delta^{13}$C and $\delta^{34}$S in Maolan area represent the composition of the typical stream waters in subtropical karst areas.

The $\delta^{13}$C value of $HCO_3^-$ in the Banzhai watershed has seasonal variation. The $\delta^{13}$C value in summer (average $-15.3‰$) is relatively negative, while that in spring (average $-12.6‰$) is relatively positive. This difference may be related to the biological activity level and rainfall intensity in different seasons. In summer, the biological activity intensity is high, the biogenic soil $CO_2$ acts on the water, forming a more negative $\delta^{13}$C value; the obvious negative correlation between the $HCO_3^-$ content and the $\delta^{13}$C value is affected by the weathering and leaching intensity under the joint action of temperature and rain.

At the Banzhai small watershed scale, $H_2SO_4$ is involved in the weathering of carbonate rocks and affects the $\delta^{13}$C value of DIC. Most of the water shows an obvious positive correlation between $SO_4^{2-}$ content in water and $\delta^{13}$C value of DIC, that is, with the increase in $SO_4^{2-}$ content, the proportion of DIC completely derived from carbonate weathering increases, so its $\delta^{13}$C value is relatively positive. When the $\delta^{34}$S in the water reveals that $SO_4^{2-}$ is more likely to come from the coal, the corresponding $\delta^{13}$C-DIC is closer to the initial value of carbonate rock, which indicates that sulfur in local coal is a driving force that cannot be ignored in the process of carbonate weathering.

**Author Contributions:** Conceptualization, methodology, software, validation, investigation, resources, writing—Original draft preparation, writing—Review and editing, visualization, supervision, project administration, funding acquisition, Y.T.; formal analysis, data curation, R.H. All authors have read and agreed to the published version of the manuscript.

**Funding:** This research received no external funding.

**Institutional Review Board Statement:** Ethical review and approval were waived for this study, due to studies not involving humans or animals.

**Informed Consent Statement:** This study does not involve humans.

**Data Availability Statement:** Data is contained within the article.

**Acknowledgments:** This work was supported jointly by National Natural Science Foundation of China (No. 41325010; 41403109).

**Conflicts of Interest:** The authors declare no conflict of interest.

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
