# Peer review of "Stable Carbon and Sulfur Isotope Characteristics of Stream Water in a Typical Karst Small Catchment, Southwest China"

_water, doi:10.3390/w13040523_

Round 1

Reviewer 1 Report

The work submitted for publication in Water has serious deficiencies in both form and content. In general, it is not planted as a scientific article, the objectives do not have any scope, the samples taken are totally insufficient to achieve significant results and the discussion of them is lacking of scientific content of a certain level. The conclusions are vain and little supported by the data obtained. From the formal point of view, although the structure is coherent, Table 1 is missing, the figures are few and of low quality. In conclusion, the work needs a lot of additional work and a better focus on the topic to be publishable.

Author Response

Reviewer 1:The work submitted for publication in Water has serious deficiencies in both form and content. In general, it is not planted as a scientific article, the objectives do not have any scope, the samples taken are totally insufficient to achieve significant results and the discussion of them is lacking of scientific content of a certain level. The conclusions are vain and little supported by the data obtained. From the formal point of view, although the structure is coherent, Table 1 is missing, the figures are few and of low quality. In conclusion, the work needs a lot of additional work and a better focus on the topic to be publishable.

Response:Thank you very much for your valuable comments on our article, pointing out many shortcomings of the article. We have revised the article according to these opinions, hoping that the results of this study can be accepted by more peers. First of all, we add Table 1 and table 2 to the appropriate position of the article. Secondly, we sort out the unclear analysis in the paper, and focus on the comparative analysis of data to supplement and improve the demonstration, so that the conclusion of the article is more logical. Although the data in this paper are not scientific discoveries of great significance, they are also in accordance with scientific laws. For example, through the analysis of the correlation between sulfur isotope and carbon isotope, we reveal that sulfur-bearing coal may be an important driving force for carbonate weathering in this area, which can also provide a theoretical supplement for the weathering of karst scholars.

Reviewer 2 Report

  The manuscript "Stable carbon and sulfur isotope characteristics of Stream water in a typical karst small catchment, southwest China" is well written and organized, but there are some issues that need to be resolved prior to its publication.
For starters, Tables are missing in the text!

Furthermore, the authors need to revise the titles of the Figures so they describe what is actually shown on them.

E.g. FIG. 5 H2SO4 involves in weathering of carbonate rocks and its influence on13C-DIC ... this is not shown on Figure 5. (same applies to all Figures).

The paper also needs a moderate revision of the language, including typos, unnecessary capital letters, etc.

Author Response

Reviewer 2:The manuscript "Stable carbon and sulfur isotope characteristics of Stream water in a typical karst small catchment, southwest China" is well written and organized, but there are some issues that need to be resolved prior to its publication.For starters, Tables are missing in the text!

Response:We have supplemented the detailed data of Table 1 and table 2 in the appropriate position of the article

Reviewer 2:Furthermore, the authors need to revise the titles of the Figures so they describe what is actually shown on them. E.g. FIG. 5 H2SO4 involves in weathering of carbonate rocks and its influence on13C-DIC ... this is not shown on Figure 5. (same applies to all Figures).

Response:We have revised the title of the figures, which can better reflect the content and connotation of the figures.

Reviewer 2:The paper also needs a moderate revision of the language, including typos, unnecessary capital letters, etc.

Response:The grammar and spelling of the article have been thoroughly checked and revised.

Reviewer 3 Report

Dear Authors

In my opinion the theme of the article is innovate and very interesting for the readers of the journal.

The authors analyzed surface water samples from the Maolan National Natural Reserved Park (MNNRP) from among ~one year, for major ion concentrations, δ13C-DIC and δ34S-SO42- to quantify the sources of solutes and chemical weathering.

It was found that HCO3- and SO42- are the main anions in the small watershed, accounting for 86.2% and 10.4% of the total anion equivalent, while Ca2+ and Mg2+ account for 76.9% and 20.5% respectively and that Mg2+ in stream water resulted mainly by dolomite weathering.

Stream water samples present the δ13C-DIC mean values of -13.92‰, lower than that of the groundwater. The δ34S-SO42- mean values was -4.35‰.

Combining δ13C-DIC and δ34S-SO42- values clearly distinguishes the principal sources of solutes, the main source is the hydrolysis of carbonate rocks and the participation of some acidic substances.

The δ13C-DIC values were negative in summer relative to that in winter, and there was a negative correlation between HCO3- content and δ13C value, suggesting the result of the interaction of temperature and precipitation intensity in different seasons.

It was found a positive correlation between SO42- content and δ13C-DIC, indicating that H2SO4 may be involved in the weathering process of carbonate rocks in small watershed scale.

The research shows concentrations of SO42- in school sampling site were higher than that of other sampling sites due to the interference from human sources. The δ34S values show that the average δ34S-SO42- in School and Liming sites are -11.1‰ and -9.12‰, respectively, suggesting a link with sulfide oxidation in underground coal seams.

The δ34S isotopic values of Village groundwater and stream water were similar to those waters of SO42- in watersheds in nearby karst areas indicating the obvious erosion of carbonate rocks by sulfuric acid.

The manuscript under revision is well structured; the language is clear. The title and abstract clearly describe the content of the manuscript. Congratulations!

In my opinion only minor revision is needed. Please find attached file.

Best regards

Author Response

Reviewer 3:In my opinion the theme of the article is innovate and very interesting for the readers of the journal.

The authors analyzed surface water samples from the Maolan National Natural Reserved Park (MNNRP) from among ~one year, for major ion concentrations, δ13C-DIC and δ34S-SO42- to quantify the sources of solutes and chemical weathering.

It was found that HCO3- and SO42- are the main anions in the small watershed, accounting for 86.2% and 10.4% of the total anion equivalent, while Ca2+ and Mg2+ account for 76.9% and 20.5% respectively and that Mg2+ in stream water resulted mainly by dolomite weathering.

Stream water samples present the δ13C-DIC mean values of -13.92‰, lower than that of the groundwater. The δ34S-SO42- mean values was -4.35‰.

Combining δ13C-DIC and δ34S-SO42- values clearly distinguishes the principal sources of solutes, the main source is the hydrolysis of carbonate rocks and the participation of some acidic substances.

The δ13C-DIC values were negative in summer relative to that in winter, and there was a negative correlation between HCO3- content and δ13C value, suggesting the result of the interaction of temperature and precipitation intensity in different seasons.

It was found a positive correlation between SO42- content and δ13C-DIC, indicating that H2SO4 may be involved in the weathering process of carbonate rocks in small watershed scale.

The research shows concentrations of SO42- in school sampling site were higher than that of other sampling sites due to the interference from human sources. The δ34S values show that the average δ34S-SO42- in School and Liming sites are -11.1‰ and -9.12‰, respectively, suggesting a link with sulfide oxidation in underground coal seams.

The δ34S isotopic values of Village groundwater and stream water were similar to those waters of SO42- in watersheds in nearby karst areas indicating the obvious erosion of carbonate rocks by sulfuric acid.

The manuscript under revision is well structured; the language is clear. The title and abstract clearly describe the content of the manuscript. Congratulations!

In my opinion only minor revision is needed. Please find attached file.

Response: Thanks to the reviewers for their positive evaluation and encouragement of this research work. We have revised some mistakes in the paper, added missing charts, revised the title of the charts, and re-analyzed the discussion on the relationship between sulfur isotope and carbon isotope of water, so as to make the conclusion revealed by the data more sufficient and scientific.

Round 2

Reviewer 1 Report

Dear authors: the manuscript has slightly improved compared to the previous original version, but it still needs important changes, some of them of a general nature and others of detail. Those of a general nature are the following:

1. The scientific objectives of the paper must be perfectly defined. Please write them appropriately, clearly and precisely, without generalizations, and in accordance with the actual scope and structure of the work.

2. The analyzed samples are still scarce (for example, there are no sewage samples from some sampling points, with which to compare the results of sulfur isotopes). With so few samples, it is not possible to draw general conclusions such as the temporal evolution of the different parameters and their causes, nor to rule out other possible causes of the origin of sulfates (for example, anthropic contamination).

3. The conclusions must be precise and respond to the previous discussion and the results obtained. In their current wording it seems that they respond to a preconceived idea of ​​the possible origin of sulfates, without a reliable demonstration.

4. The figures and tables should be improved, offering all the necessary information (see further detailed questions)

5. There are many missing references in the text that justify certain statements (see details below)

6. The text must be proofread by a native English.

  1. There are numerous small errors throughout the text that must be corrected, especially in relation to the use of capital letters.

Among the detailed comments are the following:

  1. In the Introduction chapter, sulfuric acid and nitric acid are said to be “human disturbances”. These acids are chemical products, which can be generated by natural or anthropic processes, but should not be considered a priori as such disturbances.
  2. Reference is necessary when it is said that China is the largest karst area in the world.
  3. In the Materials and Methods section, a reference is necessary when the carbonate rocks present in the area and their age are described. In addition, a much more detailed geological description is advisable, which helps to understand the relationships of water with the various lithological and mineralogical aspects.
  4. Figure 1 should include place names in the upper boxes and specify that the points in the lower box are the sampling locations.
  5. In the coordinates and altitudes of the sampling places, indicate m.a.s.l.
  6. Please, indicate the volumes of the bottles where the samples were taken.
  7. In section 3.2, some values of 13C are cited with a single decimal number, while in the rest of the work it is done with two figures. Please homogenize.
  8. Table 1 needs some improvement. I suggest putting a column with the type of sample (groundwater, specifying whether it is well or spring; surface water), also changing the date column for the "season" (spring, summer, autumn or winter); identify particularly prominent values with colors, etc.
  9. Table 2 is not well understood: What are the principal components? In addition, the information that is deduced from it must be extracted and adequately reflected in the text.
  10. In section 4.1, it is said that the pH values suggest that there is no impact of anthropic activities… Could you justify this?
  11. Similarly, groundwater is said to have a high TDS, but no data is provided.
  12. It is also said that the water in the sample from School having sulfates means that these come from “additional sources”. All this means is that, indeed, it has more sulfates than the rest of the waters, but not that the sources are different.
  13. The comparison with the waters of the Beipanjiang River is not understood, nor is it described where it is and why the comparison is made and for what ...
  14. In Figures 3, and 5 there is always a School sample that falls outside the line defined by the others. Has this sample not been considered to be in error?
  15. Citations from Telmer and Veizer (1999b) and another should be put like the rest.
  16. In section 4.3 it is stated that the 34S isotopic composition of seawater is -20 per thousand: a citation is necessary, just as when citing the presence of coal in the Banzhai basin.
  17. Figure 14 should include the values ​​(or at least the range from other work), for urban wastewater.
  18. In section 4.4 it is said that the positive correlation between sulfur and carbon isotopes necessarily implies that the weathering of carbonaceous rocks is affected by sulfuric acid: Why?
  19. The conclusions establish certain relationships between the environmental conditions of the different seasons of the year and the weathering that have not been sufficiently argued. There are not even graphs of the temporal evolution of the variables involved in the work.
  20. Finally, the hypothesis of the origin of sulfate in the waters from coal sulfides is not proven, given the possible origin of anthropic contamination.

Round 3

Reviewer 1 Report

After the corrections made, the paper is considered publishable